# Testicular Transcriptome of Males and Pseudo-Males Provides Important New Insight into Sex Reversal of *Rana dybowskii*

**DOI:** 10.3390/ani12202887

**Published:** 2022-10-21

**Authors:** Yuan Xu, Shiquan Cui, Ting Li, Zhiheng Du, Fangyong Ning, Haixu Jiang, Xiujuan Bai, Xibiao Wang, Jun Bao

**Affiliations:** 1College of Animal Science and Technology, Northeast Agricultural University, No. 600 Changjiang Road, Xiangfang District, Harbin 150030, China; 2Key Laboratory of Swine Facilities Engineering, Ministry of Agriculture and Rural Affairs, No. 600 Changjiang Road, Xiangfang District, Harbin 150030, China

**Keywords:** *Rana dybowskii*, sex reversal, pseudo-males, testicular transcriptome, *Dmrt1*

## Abstract

**Simple Summary:**

Aquaculture in Northeast China has favored *Rana dybowskii* (*R. dybowskii*) because of its regional specificity and the economic value of the female frog’s ovary, and the increasing proportion of female frogs is beneficial to improve the economic efficiency of aquaculture. Unfortunately, the sexual reversal from genotypic female to phenotypic male has led to a decrease in the proportion of females in the breeding population. In this study, differentially expressed genes between testes of sex reversal and male testes were screened by transcriptome sequencing. These findings provide a comprehensive analysis of gene expression in *R. dybowskii* sex reversal from genotypic female to phenotypic male and provide new ideas for sex control of *R. dybowskii*.

**Abstract:**

*Rana dybowskii* (*R. dybowskii*) is an ecological species found in China, Japan, Korea, and Russia. Like most amphibians, *R. dybowskii* lacks heterotypic sex chromosomes, limiting the in-depth study of sex determination and sex reversal mechanisms. Previous studies have shown that certain environmental factors can modify *R. dybowskii* genotypic females into phenotypic males, but the mechanism is still unknown. Considering the difficulties in identifying and collecting sex reversal gonads at different stages of differentiation under natural conditions, testes from sexually mature wild adult *R. dybowskii* were taken in this study, and the genotypic sex of individuals and sex reversal were identified by two male-linked genetic markers reported in our most recent findings. Transcriptome sequencing was performed on testicular tissue from males and pseudo-males, as well as female ovary tissue. The results show that the gene expression patterns of pseudo-males’ testes were similar to those of the males but highly differed from females’ ovaries. One hundred and seventeen differentially expressed genes between testes of pseudo-males and males were found, and the up-regulation of doublesex and mab-3 related transcription factor 1 (*Dmrt1)* in testes of pseudo-males may play a key role in *R. dybowskii* sex reversal.

## 1. Introduction

Sex reversal induced by myriad natural and anthropogenic changes in the environment can lead to shifts in a population’s sex ratio, with implications for population dynamics in many lower vertebrates, including frogs [1,2]. Firstly, the effects of sex reversal on population dynamics are possible in a long-term process. This point can be seen from the genotype–phenotype mismatches caused by temperature, with temperature-induced masculinization potentially leading to the loss of Y chromosomes or feminization in future generations [3,4]. Secondly, sex reversal has been found to generally have a negative effect on population size, and the extreme sex ratios caused by high sex reversal rates can significantly reduce population sizes [3]. Cotton and Wedekind [1] found that environmental sex reversal changes the predominant factor in sex determination from genetic to entirely environmental. However, populations that lose genetic sex determination may quickly go extinct if the environmental forces that cause sex reversal cease. Furthermore, several studies have shown that masculinization is generally more detrimental to populations than feminization [5,6]. Among them, White et al. [7] used a sex-specific model to assess the status of wild silverside populations and found that even slight masculinization changes in the sex ratio will reduce reproductive success. Therefore, many researchers have focused on the mechanism of frog sex reversal to solve the significant challenge of its impact on frog population dynamics [8,9,10].

*Rana dybowskii* (*R. dybowskii*) is an ecological species found in China, Japan, Korea, and Russia [11]. Even though the research studies on *R. dybowskii* are in the early stages, some of the results still show a clear sex reversal. In a previous study, Huo et al. [12] found that obvious male-biased sex ratios (greater than 7/10) existed in the wild and semi-wild populations of *R. dybowskii*. As a result of these findings, research studies have frequently focused on the influence of environmental factors (sex hormones, temperature) on *R. dybowskii* sex reversal under laboratory conditions [12,13]. However, one of the main problems with these studies is that the sex reversal of *R. dybowskii* is only inferred from population sex ratio deviations from 1:1, indicating a lack of extensive sex reversal research at the individual level, which, predictably, has become a significant obstacle in the investigation of *R. dybowskii*’s molecular mechanism of sex reversal. Subsequently, the genotypic-female-to-phenotypic-male sex reversal of *R. dybowskii* might have been implicated in obvious male-biased sex ratios that occurred in wild and semi-wild populations, as suggested by the recent evidence of sex reversal at the individual level (Figure 1) [14]. There are two characteristics of this sex reversal: (1) the sex reversal ratio of *R. dybowskii* varied geographically in various natural populations, with individuals from higher-latitude populations having the lowest ratio. (2) Under laboratory settings, the genotypic female fertilized *R. dybowskii* eggs/tadpoles growing at various temperatures had varied sex reversal ratios, which increased as the ambient temperature rose. Although two male-linked molecular markers were used to identify *R. dybowskii* sex reversal, the deeper molecular mechanisms of sex reversal remain unexplained.

It has been reported that the improper expression of genes involved in sex determination causes sex reversal and the maintenance of gonadal function in sex-reversed individuals [15,16]. The key genes involved in sex determination and sex reversal in various animals are not usually the same throughout this process [16]. Even the same gene can have different roles in different species when it comes to sex reversal. SOX9 is a critical gene in mammals for sex determination, and loss of SOX9 expression or its inappropriate activation can lead to XY female development and XX male development in humans and mice, respectively [17]. Additionally, Ge et al. [18] found that the transcription of lysine-specific demethylase 6B was elevated at a low temperature in female Trachemys scripta, which up-regulated doublesex and mab-3 related transcription factor 1 (*Dmrt1)* expression and promoted the development of male gonad. According to a study of *Cynoglossus semilaevis* by Cui et al. [19], SNPs in the *Dmrt1* gene control the transition from genotypic females to phenotypic males. However, there is a gap in the research on *R. dybowskii*’s key genes for sex determination and sex reversal.

Transcriptome sequencing has developed rapidly, and its widespread use in screening genes involved in sex determination has been already widely recognized [20,21,22]. In this study, we identified the sex reversal of *R. dybowskii* at the individual level by two male-linked molecular markers and reported an integrative gene expression analysis of testes from adult males and pseudo-males of *R. dybowskii* by transcriptome sequencing. We aimed to screen the genes related to sex reversal of *R. dybowskii* via this method and then determine the functional regions that can affect sex reversal of *R. dybowskii*, as well as to provide a theoretical basis for resource conservation of *R. dybowskii*.

## 2. Materials and Methods

### 2.1. Animals and Tissue Sampling

Twenty-five *R. dybowskii* males and five females were collected from Xinglong, Heilongjiang province. The frogs were euthanized in 0.2% MS222, and their muscles were then collected and preserved in 75% ethanol at −20 °C for DNA extraction. Testes or ovaries were collected from each frog, followed by preservation in liquid nitrogen for RNA extraction.

### 2.2. DNA Extraction and Identification of Genotypic Sex

DNA of muscle was extracted using the method described by Xu et al. [23]. Identification of genotypic sex of *R. dybowskii* was performed using PCR amplification, based on two markers screened in our recent study with two pairs of primers (Combination-1: forward primer 5′GGCTATTCGTCGCTACTAAAGG3′/reverse primer 5′GACTGCGTACGAATTTGA3′; Combination-2: forward primer 5′GGTCATTCCTTGTCCTAATTATCAG3′T/reverse primer 5′GACTGCGTACGAATTGAG3′) [14]. PCR products were analyzed by electrophoresis in 14% polyacrylamide gel.

### 2.3. RNA Extraction

According to the results of genotypic sex identification, the twenty-five phenotypic males with the two male-specific fragments were considered to be males, while the males without the two fragments were considered to be pseudo-males. Total RNA was extracted from the gonadal tissues of males, pseudo-males, and females using TIANGEN RNA extraction kits (TIANGEN Technology, Beijing, China). The extracted RNA was electrophoresed on 1.2% agarose gels to ensure the integrity of RNA and no DNA contamination. Precise quantification of RNA concentration was performed using Qubit2.0 Fluorometer (SEM), USA, and Agilent 2100 Bioanalyzer (Nanodrop, Wilmington, DE, USA) was used to accurately detect RNA integrity.

### 2.4. cDNA Library Preparation and Illumina Sequencing

Seven cDNA libraries were constructed using a TruSeq RNA Sample Preparation Kit (Illumina, San Diego, CA, USA). Three libraries (XLDX155, XLDX159, and XLDX161) from the male testis group (MT), three libraries (XLDX162, XLDX172, and XLDX174) from the pseudo-male testis group (PMT), and one library (XLDC57, XLDC76, and XLDC78) from the mixed female ovary group (FO) were preliminarily quantified by qubit2.0 fluorometer (Thermo Fisher, PA, USA). The libraries were then diluted to 1.5 ng μL^−1^. The insert size of the library was detected using Agilent 2100 Bioanalyzer. After passing the library inspection, Frasergen Information Co., Ltd. (Wuhan, China) was instructed to use Illumina hiseq 4000 for sequencing.

### 2.5. Sequence Data Processing and Analysis

To ensure the quality and reliability of data analysis, it is necessary to filter the original data by removing reads with adapter, reads containing n (n indicates that the base information cannot be determined), and low-quality reads (base number of Qphred ≤ 20 accounting for more than 50% of the whole read length) to obtain clean reads. Meanwhile, Q20, Q30, and GC contents of clean data were calculated. All subsequent analyses were based on clean data. Trinity software (http://trinityrnaseq.sourceforge.net/ accessed on 21 June 2019) was used for de novo assembly of transcriptome data to obtain the UniGene sequence. The assembled transcript information was stored in FASTA format. Diamond (version 0.8.18) was used to compare the transcript with the NR database, Swiss-Prot, and KO database (similarity > 30%, E-value < 1 × 10^−5^). NCBI blast 2.2.29 + and KOBAS 3.0 were used to compare transcripts with the KOG database and KEGG database, respectively.

### 2.6. Identification of Differentially Expressed Genes (DEGs)

The gene expression was calculated by fragments per kilobase of transcript per million mapped reads (FPKM). DEGs were mainly screened out between (i) male testis group and pseudo-male testis group, and they were similarly screened out between (ii) male testis group and female ovary group and (iii) pseudo-male testis group and female ovary group. Benjamin Hochberg’s method was used to screen differentially expressed genes [24]. The screening threshold was false discovery rate (FDR) < 0.05, log2fc (fold change (condition 2/condition 1) for a gene) > 1, or log2fc < −1.

In organisms, different genes coordinate with each other to perform their biological functions. For DEGs in this study, GO and KEGG enrichment analyses were performed using GOseq software and KOBAS software, respectively.

### 2.7. Validation of DEGs

To further validate the reliability of transcriptomic data, qRT-PCR was performed. Three genes, *Dmrt1*, 3-beta-hydroxysteroid dehydrogenase (*3β-HSD*), and cytochrome P450 26B1 (*CYP26B1*), related to male/female sex differentiation were selected for qRT-PCR analysis. The housekeeping gene β-actin was used as an internal normalization control, and the specific primers for the candidate genes used for qPCR were designed by Primer Premier 5.0 (Appendix A). RNA (about 500 ng) samples were measured and treated with RQ1 RNase-Free DNase (Promega, Madison, WI, USA) to remove genomic DNA. The cDNA was synthesized using a reverse transcriptase reagent kit (PrimeScript™ RT reagent Kit, Takara, Japan)). The qPCR was performed with SYBR Premix Ex Taq II (Takara, Japan). The reactions were carried out in a total volume of 25 μL, containing 2.5 μL of diluted cDNA, 2.5 μL of each primer, and 12.5 μL of SYBR Green PCR Master Mix, with the following cycling profile: 95 °C for 15 min for polymerase activation, followed by 40 cycles at 95 °C for 15 s, at 55 °C for 30 s, and at 70 °C for 30 s. Each sample was processed in triplicate in the Roche LightCycler 480 Real-Time PCR System (Roche). All data were analyzed using the 2^−∆∆Ct^ method. All data were analyzed using SPSS 19.0, and *p* value was calculated using Student’s *t*-test. Values of *p* < 0.05 were considered to indicate a statistically significant difference, and those of *p* < 0.01 were considered to indicate a statistically extremely significant difference.

## 3. Results

### 3.1. Identification of Genotypic Sex

Two pairs of primers were applied to identify the genotypic sex in the population of *R. dybowskii*. As shown in Figure 2a, when the primer combination-1 was used to identify the genotypic sex of 25 males and 5 females, a 222 bp male-specific fragment was observed in 13 out of 25 males, which was completely absent in all 5 females. For the primer Combination-2, a 261 bp long fragment was amplified in 12 males and absent in all females (Figure 2b). Phenotypic males with the two fragments were considered to be males, while the males without the two fragments were considered to be pseudo-males. The gonads of three males (testis), three pseudo-males (testis), and three females (ovary) were chosen for transcriptome sequencing based on the results of genotypic sex identification (Table 1).

### 3.2. Transcriptome Sequencing and Assembly

Through transcriptome sequencing of *R. dybowskii* gonads, 166,881,915 paired-end reads were obtained (Table 2). After assembling and splicing, a total of 159,757 transcripts (>500 bp) were generated, with an average length of 874 bp, GC content of 46.20%, and N50 length of 1486 bp, indicating higher integrity of assembly in our study. The summary of the transcriptome sequencing data is shown in Table 3.

### 3.3. Functional Annotation and Classification

Comparing the 159,757 assembled transcripts with KOG, KEGG, NR, GO and Swiss-Prot databases, the annotation results show that 48,364 transcripts (30.27%) were annotated to the NR database and 39,064 transcripts (24.45%) were annotated to the Swiss-Prot database (Table 4).

### 3.4. Analyses of Differentially Expressed Genes

To reveal the changes in gene expression at the transcriptome level in sex reversal gonad, and to characterize the partial molecular mechanism of sex reversal in *R. dybowskii*, three groups of digital gene expression libraries were constructed. The first came from the testis of MT, the second from the testis of PMT, and the third from the ovary of FO. The R language package deseq was used for different analyses. The screening threshold was FDR (false discovery rate) < 0.05, log2fc (fold change (condition 2/condition 1) for a gene) > 1, or log2fc < −1.The DEGs for MT vs. PMT, MT vs. FO, and PMT vs. FO are shown in Table 5.

(i) Compared with MT, the number of up (down)-regulated transcripts in the PMT library was 81 (36), and it was 2894 (3031) in the FO library. Among them, the libraries of PMT and FO had 16 similar genes that were down-regulated, however, no up-regulated genes were observed. Interestingly, we found that among the up-regulated genes in pseudo-male testis, three genes (double sex and mab-3 related transcription factor 1, cytochrome P450 26B1, and cocaine- and amphetamine-regulated transcription protein-like) were not expressed in female ovary. (ii) Compared with PMT, the number of up (down)-regulated transcripts in the MT library was 36 (81), and it was 2958 (2938) in the FO library; Among them, the libraries of MT and FO have the same 45 genes that are up-regulated and 3 similar genes that are down-regulated, one of which may be Von Willebrand factor, a domain containing protein 5A-like. (iii) Compared with the FO library, the number of up (down)-regulated transcripts in the PMT library was 2938 (2958), and it was 3031 (2894) in the MT library; Among them, 1886 genes were similar and up-regulated (including 832 genes not expressed in the ovary), and 2150 genes were similar and down-regulated (159 genes not expressed in testis).

### 3.5. GO Analysis of DEGs

GO annotations were used to assess the possible functions of DEGs. The up-regulated and down-regulated DEGs enriched 171 and 135 GO terms, respectively (*p* < 0.05) (Appendix A). The major enriched GO terms for up- and down-regulated DEGs of MT-vs.-PMT are shown in Figure 3a,b, respectively. In the up-regulated DEGs, single-organism process (GO:0044699), cell communication (GO:0007154), response to stress (GO:0006950), chemical homeostasis (GO:0048878), homeostatic process (GO:0042592), response to external stimulus (GO:0009605), and positive regulation of response to stimulus (GO:0048584) were the seven terms enriched with the most genes, with 19, 9, 6, 4, 4, 4, and 4 genes, respectively. Accordingly, organonitrogen compound metabolic process (GO:1901564), nitrogen compound metabolic process (GO:0006807), primary metabolic process (GO:0044238), organic substance metabolic process (GO:0071704), protein metabolic process (GO:0019538), and proteolysis (GO:0006508) were the six terms enriched with the most genes in the down-regulated DEGs, with 4, 4, 4, 4, 3, and 2 genes, respectively.

The up- and down-regulated DEGs of FO-vs.-MT enriched 696 and 452 GO terms (*p* < 0.05), respectively (Appendix A). The top 30 enriched GO terms for up- and down-regulated DEGs of FO-vs.-MT are shown in Figure 3c,d, respectively. In the up-regulated DEGs, single-organism process (GO:0044699), organonitrogen compound metabolic process (GO:1901564), protein metabolic process (GO:0019538), multicellular organismal process (GO:0032501), and cellular protein metabolic process (GO:0044267) were enriched with 452, 258, 223, 203, and 186 genes, respectively. Accordingly, cell (GO:0005623), cell part (GO:0044464), intracellular (GO:0005622), intracellular part (GO:0044424), and organelle (GO:0043226) were the five terms enriched with the most genes in the down-regulated DEGs, with 910, 909, 857, 824, and 753 genes, respectively.

The up- and down-regulated DEGs of FO-vs.-PMT enriched 865 and 439 GO terms (*p* < 0.05), respectively (Appendix A). The top 30 enriched GO terms for up- and down-regulated DEGs of FO-vs.-PMT are shown in Figure 3e,f, respectively. In the up-regulated DEGs, single-organism process (GO:0044699), biological regulation (GO:0065007), regulation of biological process (GO:0050789), regulation of cellular process (GO:0050794), and membrane (GO:0016020) were the five terms enriched with the most genes, with 447, 352, 333, 314, and 313 genes, respectively. Accordingly, intracellular (GO:0005622), intracellular part (GO:0044424), organelle (GO:0043226), intracellular organelle (GO:0043229), and membrane-bounded organelle (GO:0043227) were the five terms enriched with the most genes in the down-regulated DEGs, with 760, 727, 663, 631, and 605 genes involved, respectively.

### 3.6. KEGG Analysis of DEGs

To understand the molecular mechanisms of sex reversal from genotypic female to phenotypic male, KEGG was used to analyze the function of DEGs. The up-regulated DEGs with KEGG annotations for MT-vs.-PMT, FO-vs.-MT, and FO-vs.-PMT were enriched to 8, 20, and 20 pathways (*p* < 0.05), while the down-regulated DEGs were enriched to 4, 20, and 20 pathways (*p* < 0.05) (Figure 4).

In the KEGGs enrichment of up-regulated DEGs in MT-vs.-PMT, phagosome (ko04145), TGF-beta signaling pathway (ko04350), HIF-1 signaling pathway (ko04066), cell adhesion molecules (ko04514), and leukocyte transendothelial migration (ko04670) were the five pathways enriched with the most genes, with two genes each. In the KEGGs enrichment of down-regulated DEGs in MT-vs.-PMT, butanoate metabolism (ko00650), necroptosis (ko04217), and NOD-like receptor signaling pathway (ko04621) were the four pathways enriched with the most genes, with one gene each.

In the KEGGs enrichment of up-regulated DEGs in FO-vs.-MT, ribosome (ko03010), vascular smooth muscle contraction (ko04270), inflammatory mediator regulation of TRP channels (ko04750), autophagy—animal (ko04140), and neurotrophin signaling pathway (ko04722) were the five pathways enriched with the most genes, with 80, 37, 31, 29, and 25 genes, respectively. In the KEGGs of down-regulated DEGs in FO-vs.-MT, cell cycle (ko04110), purine metabolism (ko00230), pyrimidine metabolism (ko00240), cell cycle—yeast (ko04111), and oocyte meiosis (ko04114) were the five pathways enriched with the most genes, with 44, 35, 29, 28, and 27 genes, respectively.

In the KEGGs of up-regulated DEGs in FO-vs.-PMT, vascular smooth muscle contraction (ko04270), inflammatory mediator regulation of TRP channels (ko04750), autophagy—animal (ko04140), GnRH signaling pathway (ko04912), and cAMP signaling pathway (ko04024) were the five pathways enriched with the most genes, with 37, 33, 27, 26, and 26 genes, respectively.

The five pathways enriched with the most genes in the KEGGs enrichment of down-regulated DEGs in PMT-vs.-FO were cell cycle (ko04110), purine metabolism (ko00230), pyridine metabolism (ko00240), cell cycle—yeast (ko04111), and RIG-I-like receptor signaling pathway (ko04622), with 43, 39, 32, 29, and 29 genes, respectively.

### 3.7. Change in Gene Expression Related to Sex Determination

Changes in gene expression levels relevant to sex determination were determined using transcriptome data to better understand the possible molecular mechanism of *R. dybowskii* sex reversal. Among 117 DEGs of MT-vs.-PMT, *Dmrt1* was the only gene related to male sex determination and enriched four GO terms, including male sex determination (GO:0030238), *p* = 0.0045; sex determination (GO:0007530), *p* = 0.016; development of primary male sexual characteristics (GO:0046546), *p* = 0.0625; and male sex differentiation (GO:0046661), *p* = 0.0733. The expression of *Dmrt1* genes was increased in PMT but not found in FO.

In 117 DEGs of MT-vs.-PMT, there were 26 other up-regulated DEGs in PMT, such as *Dmrt1*, that hit against the GO terms, and they were 3-beta-hydroxysteroid dehydrogenase, hypothetical protein, alpha-mannosidase 2, cAMP-specific 3’,5’-cyclic phosphodiesterase 4B, carbohydrate sulfotransferase 10, CD99 antigen-like, claudin-4-like, cocaine- and amphetamine-regulated transcript protein-like, cytochrome P450 26B1, follistatin, histone H1.0-B, interferon-induced helicase C domain-containing protein 1, large neutral amino acids transporter, phosphoserine phosphatase, pleckstrin homology domain-containing family F member 1, radical S-adenosyl methionine domain-containing protein 2, RNA-binding motif, single-stranded interacting protein 1, serine/threonine-protein kinase pim-3-like, solute carrier family 2, facilitated glucose transporter member 1-like, synembryn-A, TNFAIP3-interacting protein 1, TNFAIP3-interacting protein 2, transferrin receptor protein 1, ubiquitin carboxyl-terminal hydrolase 28, vesicle-associated membrane protein 8, and WW domain binding protein 1-like. In addition, mucin-5AC-like, pulmonary surfactant-associated protein B, and chymotrypsinogen A-like were the down-regulated DEGs in PMT that hit against the GO terms.

### 3.8. Verification of Transcriptome Data by qRT-PCR

To further validate the transcriptomic data, the three genes *Dmrt1*, *3β-HSD*, and *CYP26B1* were selected for qRT-PCR analysis. The gene expression patterns revealed by qRT-PCR analysis were similar to the RNA-Seq results (Figure 5), confirmed by the fact that mRNA expression levels of *Dmrt1*, *3β-HSD*, and *CYP26B1* were significantly up-regulated in testes of pseudo-males, supporting the reliability of the RNA-Seq data.

## 4. Discussion

Sex-specific molecular markers are being used to identify genotypic sex in an increasing variety of frogs, particularly those without heteromorphic sex chromosomes [25,26]. In this work, two male-specific molecular markers were used to determine the genotypic sex of *R. dybowskii*, and the findings suggest that the amplification of two male-specific molecular markers was inconsistent in just one phenotypic male individual. A similar phenomenon (this kind of inconsistency) has been seen in *R. temporaria*, *R. dalmatina*, and *R. dybowskii* and is thought to be caused by sex chromosomal recombination throughout evolution [14,26,27,28,29]. To ensure the accuracy of genotypic sex identification, the consistency of two amplification results in an individual was used to assess genotypic sex, which meant that phenotypic male individuals that amplified both male-specific molecular markers were considered genotypic male, and phenotypic male individuals that lacked both male-specific molecular markers were considered genotypic female. In contrast to the failure of chromosomal karyotype analysis and Ag-band to identify *R. dybowskii* genotypic sex [30,31], sex-specific molecular markers proved to be an efficient and easy technique to identify *R. dybowskii* genotypic sex in our study.

Sex reversal is a mismatch between genotypic and phenotypic sex [26,29,32]. We found no sex reversal from genotypic male to phenotypic female in our wild population based on whether the phenotypic sex matched the genotypic sex, which is consistent with earlier studies [14]. The reason for this could be that wild rearing conditions, in which the environmental factors were likely not optimal for sex reversal from genotypic male to phenotypic female, inhibited this kind of sex reversal. When it comes to temperature-induced sex reversal in vertebrates, high temperatures promote the induction of phenotypic males in fish and amphibians, whereas low temperatures do not always result in female sex reversal [33]. In general, frog sex reversal from genotypic male to phenotypic female has only been observed in laboratory estrogen induction trials [34], which is uncommon in wild populations. In our study, the pseudo-males, which were phenotypically male and genotypically female, were distinguished from males, providing excellent material for exposing the molecular mechanism of sex reversal in *R. dybowskii* at the individual level.

In order to understand the molecular mechanism of sex reversal in *R. dybowskii*, transcriptome analysis was performed, and only 117 DEGs were screened between male and pseudo-male testes, which is much lower than the number of DEGs screened from testis and ovary in similar studies [35,36]. DEGs up to 35539 were found between the testes and ovaries of 15-month-old *Hoplobatrachus rugulosus*, an endangered frog from southern China [36], and a total of 13,448 DEGs were found between stage III ovaries and testes of Trachinotus ovatus [35]. Ye et al. [37] screened 1893 DEGs at the transcript level between the ovaries of females and the testes of pseudo-males in the sex reversal of *Cynoglossus semilaevis*. Although we only selected one mixed ovary of *R. dybowskii* as the control, the result could roughly reflect that the 5896 DEGs between ovaries of females and testes of pseudo-males make it more difficult to effectively screen the genes involved in sex reversal in *R. dybowskii*. For this reason, sex reversal gonads were considered as important resources for studying the molecular process of sex reversal to address the issue. In comparison to 5434 DEGs between testes of males and ovaries of females, Wang et al. [21] correctly screened 5 DEGs between testes of males and pseudo-males, drastically reducing the range of candidate genes involved in sex reversal of *Nile tilapia*. In this research, we could only use adult male and pseudo-male testes as the major study objects, since phenotypic sex and sex reversal cannot be determined until stage G43 [13,38]. In our research, 117 DEGs from testes of males and pseudo-males enriched 171 GO terms including the single-organism process, cell communication, response to stress, chemical homeostasis, and the homeostatic process. 117 DEGs from testis of male pseudo-male enriched 6 KEGG pathways, including the phagosome, cell adhesion molecules, HIF-1 signaling pathway, TGF-beta signaling pathway, and leukocyte transendothelial migration. Our results defined the target candidate gene related to the sex reversal of *R. dybowskii* in a relatively small range.

Among 117 DEGs of MT-vs.-PMT, *Dmrt1*, the sole gene related to male sex determination, was not expressed in the ovary of female *R. dybowskii*. The fundamental explanation for this is that *Dmrt1* is expressed in tissue specificity and at distinct stages of gonad development. *Dmrt1* was exclusively found in the testis of *R. rugosa*, *R. chensinensis*, and *Scatophagus argus*, but not in the ovary and other organs of these species [39,40,41]. Even though *Dmrt1* was found in the ovary of teleost fish (*Schizothorax kozlovi*), the expression was 17-fold greater in the testis than in the ovary [42]. In *R. rugosa* and *R. chensinensis*, the *Dmrt1* gene was initially expressed in the developing testis at stages G25 and G30, respectively, and the expression became increased from these stages [39,40]. *Dmrt1* was screened in this study by transcriptome sequencing between male and pseudo-male testes (sex reversal), indicating that *Dmrt1* may play a critical role in the process of sex reversal or in the maintenance of sex reversal.

By comparing male testis transcriptomes, we found that *Dmrt1* was up-regulated in pseudo-male *R. dybowskii*, which contradicted the expression pattern of *Dmrt1* in male and pseudo-male fish testes. Female ovaries have considerably lower mRNA levels of *Dmrt1* than male testes, and there is no difference between testes of males and pseudo-males in *Nile tilapia* [21]. Additionally, Wang et al. [22] reported that the relative expression of *Dmrt1* was much higher in the testis of male Tongue Sole (*Cynoglossus semilaevis*) than in pseudo-males. This discrepancy in expression patterns of *Dmrt1* of male and pseudo-males among species may be due to species-specific variations in sex determination processes. In *R. dybowskii*, *Dmrt1* expression was probably dose-compensated in pseudo-males by up-regulation to a level, which ensures the development of testis and maintenance of reproductive function consistent with males. This process may be related to testosterone synthesis, which can be seen from up-regulated expression of genes such as *3β-HSD* and *CYP26B1* in pseudo-male testis. However, determining how these genes regulate and interact in this process of sex reversal requires further research.

## 5. Conclusions

In conclusion, we screened for the first time the key genes involved in *R. dybowskii* sex reversal at the transcriptome level. Between the testes of males and pseudo-males, 117 DEGs were detected that were related to single-organism processes, as well as cell communication, response to stress, chemical homeostasis, and homeostatic processes. In addition, 171 GO terms and 41 KEGG pathways were extracted. The results imply that *R. dybowskii* undergoes and maintains sex reversal in response to environmental stress by up-regulating *Dmrt1*. The genes and pathways identified will facilitate future research into the molecular mechanisms of *R. dybowskii* sex reversal.

## Figures and Tables

**Figure 1 animals-12-02887-f001:**
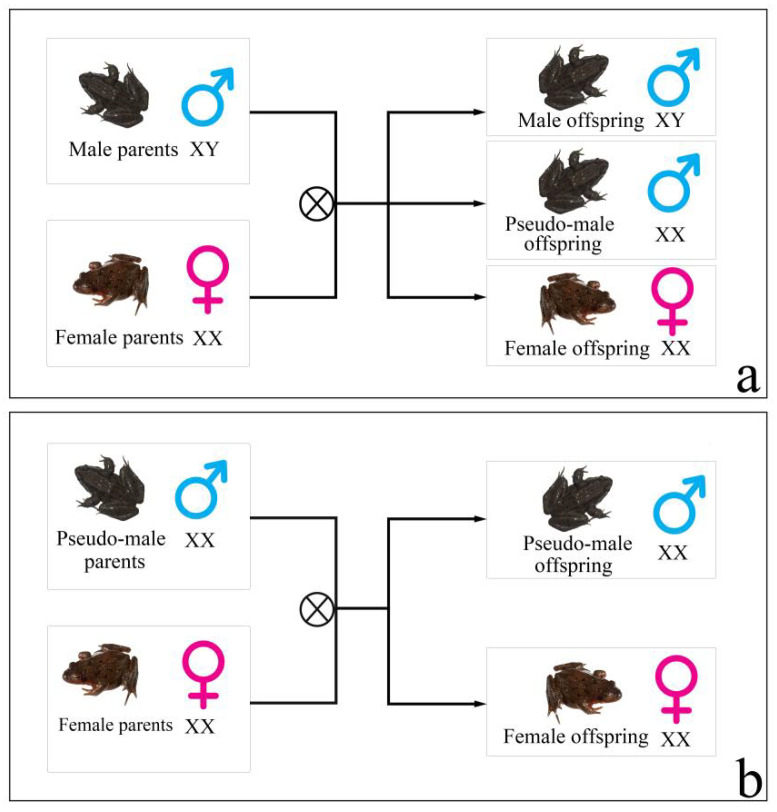
The offspring of male and pseudo-male *R. dybowskii* parents. (**a**) The offspring of male. (**b**) The offspring of pseudo-male.

**Figure 2 animals-12-02887-f002:**
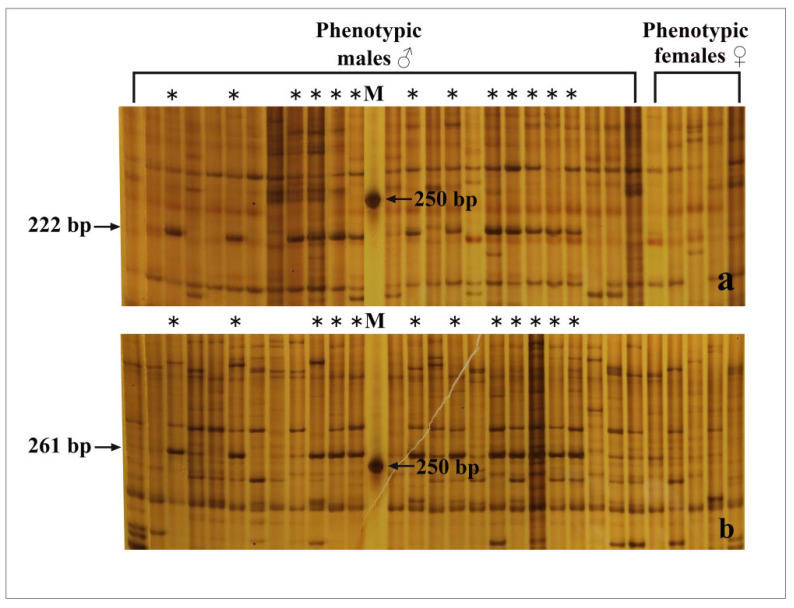
Polyacrylamide gel electrophoresis (PAGE) images of *R. dybowskii* amplified with two primer combinations. (**a**) Male-linked molecular marker (222 bp) obtained with the primer combination-1. (**b**) Male-linked molecular marker (261 bp) obtained with the primer combination-2. The individuals with ‘*’ were amplified with the male-specific markers.

**Figure 3 animals-12-02887-f003:**
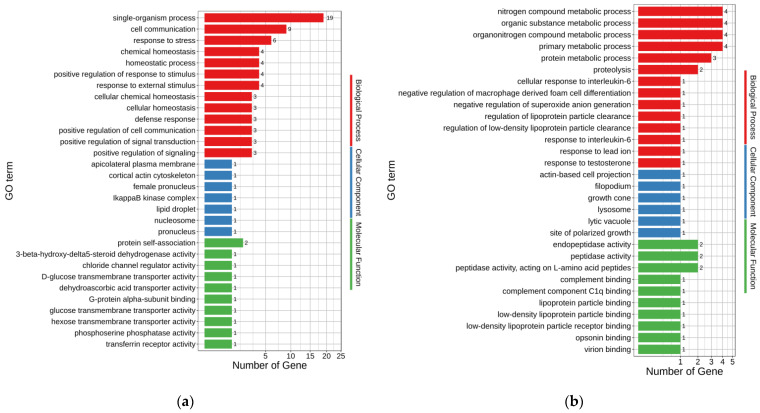
Top 30 enriched GO terms for DEGs. (**a**) Up-regulated DEGs of MT-vs.-PMT. (**b**) Down-regulated DEGs of MT-vs.-PMT, (**c**) Up-regulated DEGs of FO-vs.-MT. (**d**) Down-regulated DEGs of FO-vs.-MT. (**e**) Up-regulated DEGs of FO-vs.-PMT. (**f**) Down-regulated DEGs of FO-vs.-PMT.

**Figure 4 animals-12-02887-f004:**
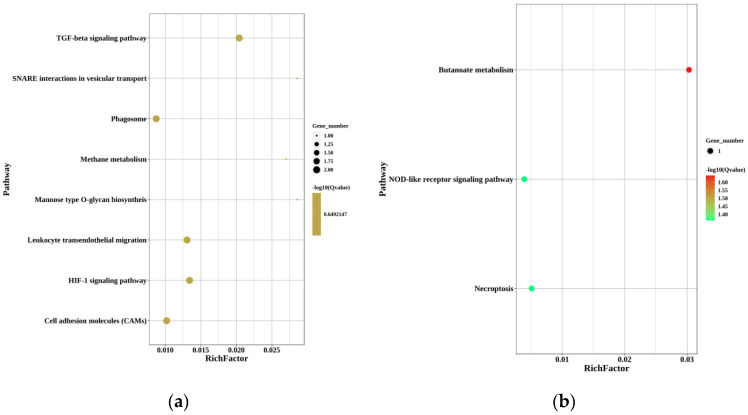
Enriched KEGG pathways for DEGs of MT-vs.-PMT, FO-vs.-MT, and FO-vs.-PMT. (**a**) Up-regulated DEGs of MT-vs.-PMT. (**b**) Down-regulated DEGs of MT-vs.-PMT. (**c**) Up-regulated DEGs of FO-vs.-MT. (**d**) Down-regulated DEGs of FO-vs.-MT. (**e**) Up-regulated DEGs of FO-vs.-PMT. (**f**) Down-regulated DEGs of FO-vs.-PMT.

**Figure 5 animals-12-02887-f005:**
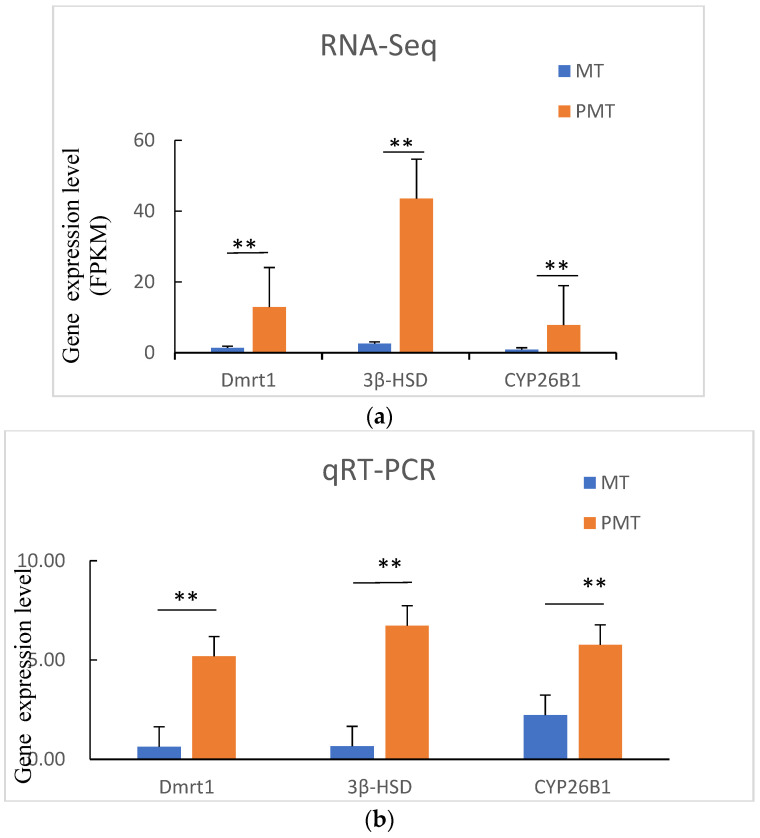
Validation of RNA-Seq results using RT–qPCR. The transcript expression levels of the selected genes were normalized to that of the β-actin gene. (**a**) The relative expression level of DEGs in MT-vs.-PMT groups determined by RNA-seq. (**b**) The relative expression level of DEGs in MT-vs.-PMT groups determined by qRT-PCR. ** *p* < 0.01.

**Table 1 animals-12-02887-t001:** *R. dybowskii* individuals were selected for transcriptome sequencing.

Sample Code	Phenotypic Sex	Genotypic Sex	Sex Reversal	Group
XLDC57	♀	female	No	FO
XLDC76	♀	female	No	FO
XLDC78	♀	female	No	FO
XLDX155	♂	male	No	MT
XLDX159	♂	male	No	MT
XLDX161	♂	male	No	MT
XLDX162	♂	female	Yes (pseudo-male)	PMT
XLDX172	♂	female	Yes (pseudo-male)	PMT
XLDX174	♂	female	Yes (pseudo-male)	PMT

**Table 2 animals-12-02887-t002:** Quality assessment of the sequencing data.

Sample	Clean Reads Pairs	Clean Base (bp)	Length	Q30 (%)	GC (%)
XLDC5776	24,095,689	7,228,706,700	150; 150	93.5; 91.7	46.6; 46.7
XLDX155	20,644,854	6,193,456,200	150; 150	94.2; 92.9	46.7; 46.8
XLDX159	22,581,904	6,774,571,200	150; 150	93.8; 92.9	46.8; 46.9
XLDX161	21,669,055	6,500,716,500	150; 150	93.7; 92.0	46.0; 46.1
XLDX162	22,166,479	6,649,943,700	150; 150	93.9; 93.0	45.3; 45.4
XLDX172	23,827,834	7,148,350,200	150; 150	93.9; 92.9	46.0; 46.0
XLDX174	31,896,100	9,568,830,000	150; 150	93.7; 92.8	45.4; 45.5

**Table 3 animals-12-02887-t003:** Frequency distribution of assembly length.

Transcript Length Interval	<500 bp	500–1 k	1 k–2 k	>2 k	Total	GC(%)
Number of transcripts	83,118	36,762	23,314	16,563	159,757	45.4; 45.5

**Table 4 animals-12-02887-t004:** Result of annotation.

Database	Number of Transcripts (Percentage)
Total	159,757 (100%)
KOG	17,467 (10.93%)
KEGG	27,321 (17.10%)
NR	48,364 (30.27%)
GO	21,321 (13.35%)
Swiss-Prot	39,064 (24.45%)
Unknown	111,026 (69.50%)

**Table 5 animals-12-02887-t005:** Differentially expressed genes among the gonadal tissues of males, pseudo-males, and females.

Group	Total	Up-Regulated	Down-Regulated	Q20 (%)
MT-vs.-PMT	117	81	36	MT-vs.-PMT
MT-vs.-FO	5925	2894	3031	MT-vs.-FO
PMT-vs.-MT	117	36	81	PMT-vs.-MT
PMT-vs.-FO	5896	2958	2938	PMT-vs.-FO
FO-vs.-PMT	5896	2938	2958	FO-vs.-PMT
FO-vs.-MT	5925	3031	2894	FO-vs.-MT

## Data Availability

The data presented in this study are available on request. These data are not publicly available to preserve the data privacy of the commercial farm.

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
