# Peer review of "Testicular Transcriptome of Males and Pseudo-Males Provides Important New Insight into Sex Reversal of Rana dybowskii"

_animals, 2022, doi:10.3390/ani12202887_

Round 1
Reviewer 1 Report
Comments for the authors
In this study, the authors focused on Dybowski’s frog, Rana dybowskii, and used transcriptome sequencing to identify 117 differentially expressed genes between sex reversal and male testis, and then extracted 171 GO terms and 41 KEGG pathways. Furthermore, the authors discovered that Dmrt1, an important gene for the development of male gonads in several vertebrates, increased in the testis of the genotypic female pseud male. This is a critical discovery. The findings of this study can be used to investigate the environmental factors that change genotypic female frogs into phenotypic males, as well as the mechanism of sex reversal.
The methodology is sound, and the manuscript is well-written. Essentially the reviewer recommends that this manuscript be published. The following suggestions would help to enhance this manuscript.
Comments
1. Dmrt1 expression is only found in the testis of Rana rugosa, but not in the ovary. Additionally, testosterone injection induces the development of XX sex-reversed gonads as well as the expression of the Dmrt1 gene in the gonads (Shibata et al., 2002, cited in this manuscript: reference number 37). These findings suggest that testosterone is a key factor in the reversal of sex from genotypic females to phenotypic males. The reviewer suggests that the authors discuss the possible role of testosterone in sex reversal.
2. 3-beta-hydroxysteroid dehydrogenase is mentioned in the Results Section 3.7. This is a key steroidogenic enzyme in the production of sex steroid hormones such as testosterone. In response to the previous comment, if there is anything in the data, the authors should discuss the effect of steroidogenesis on sex reversal.
Author Response
Response to Reviewer 1 Comments
Point 1: Dmrt1 expression is only found in the testis of Rana rugosa, but not in the ovary. Additionally, testosterone injection induces the development of XX sex-reversed gonads as well as the expression of the Dmrt1 gene in the gonads (Shibata et al., 2002, cited in this manuscript: reference number 37). These findings suggest that testosterone is a key factor in the reversal of sex from genotypic females to phenotypic males. The reviewer suggests that the authors discuss the possible role of testosterone in sex reversal.
Response 1: Thank you very much for your valuable suggestion. We described the possible role of testosterone in sex reversal (marked in red Line 446-452).
Point 2: 3-beta-hydroxysteroid dehydrogenase is mentioned in the Results Section 3.7. This is a key steroidogenic enzyme in the production of sex steroid hormones such as testosterone. In response to the previous comment, if there is anything in the data, the authors should discuss the effect of steroidogenesis on sex reversal.
Response 2: Thank you very much for your valuable suggestion. We briefly discussed the effect of steroidogenesis on sex reversal (marked in red in Line 343-358, and Line 446-452).
Due to lack of determination of hormone content, the effect of steroidogenesis on sex reversal needs further research.

Reviewer 2 Report
This manuscript by Xu et al. generated the differentially expressed genes between testis of sex reversal and male testis screened by transcriptome sequencing. These results provided a novel approach for the identification and characterization of the key genes in sex reversal Rana dybowskii. However, there is no experimental verification about these DEGs and conclusions are based on the published literature and speculation. Additionally, there are other errors in the manuscript.
1. L38:do not start sentences with numbers.
2. L40: “Dmrt1” gene names need to be italicized. “the upregulation of Dmrt1” need to demonstrate which tissue are compared to up-regulate expression.
3. L64: “R. dybowskii”. L360: “Hoplobatrachus rugulosus”. Latin names are italicized. Please check through the full manuscript.
4. L83: there is one more word “of” here. Please check again
5. In the Materials and Methods, please indicate the origin of the pseudomale used for transcriptome sequencing.
6. In the Table 5, the column of Q20 is redundant. Please check again. Actually, the information this table would be better to use histogram.
7. In the Figure 3, you do not need to show the top 50 GO terms, and the words in the Fig. 3 are too small to read. Just show top 20 or 30 GO terms. You pay attention on the significantly-enriched GO terms.
8. In the Figure 4, you can integrate the figure of KEGG pathway into one figure.
9. In the 3.7 change in gene expression related to sex determination, you can not only pat attention to the dmrt1 gene, but also the genes with similar expression trends to the dmrt1. It can also enrich the content of the discussion section.
Author Response
Response to Reviewer 2 Comments
Point 1: There is no experimental verification about these DEGs and conclusions are based on the published literature and speculation.
Response 1: Thank you very much for your valuable suggestion. We added the verification about these DEGs in “2. Materials and Methods” and “3. Results” (marked in red in Line 171-186, and Line 343-358).
Point 2: L38:do not start sentences with numbers.
Response 2: Thank you very much for your valuable suggestion. We changed the “117” to “One hundred and seventeen” (marked in red in Line 39).
Point 3: L40: “Dmrt1” gene names need to be italicized. “the upregulation of Dmrt1” need to demonstrate which tissue are compared to up-regulate expression.
Response 3: Thank you very much for your valuable suggestion. We italicized the “Dmrt1”, and accurately described “ up-regulate expression“ (marked in red in Line 40-41) .
Point 4: L64: “R. dybowskii”. L360: “Hoplobatrachus rugulosus”. Latin names are italicized. Please check through the full manuscript.
Response 4: Thank you very much for your valuable suggestion. We changed the Latin names in our paper to italicized (marked in red).
Point 5: L83: there is one more word “of” here. Please check again.
Response 5: Thank you very much for your valuable suggestion. We deleted the redundant “of”.
Point 6: In the Materials and Methods, please indicate the origin of the pseudomale used for transcriptome sequencing.
Response 6: Thank you very much for your valuable suggestion. We added the the origin of the pseudomale used for transcriptome sequencing (marked in red in Line 129-131).
Point 7: In the Table 5, the column of Q20 is redundant. Please check again. Actually, the information this table would be better to use histogram.
Response 7: Thank you very much for your valuable suggestion. We deleted the column of Q20 in the Table 5.
Point 8: In the Figure 3, you do not need to show the top 50 GO terms, and the words in the Fig. 3 are too small to read. Just show top 20 or 30 GO terms. You pay attention on the significantly-enriched GO terms.
Response 8: Thank you very much for your valuable suggestion. We changed the number of significantly-enriched GO terms to 30 (in Figure 3).
Point 9: In the Figure 4, you can integrate the figure of KEGG pathway into one figure.
Response 9: Thank you very much for your valuable suggestion. We sought the help of the data analyst, who suggested that the figure of KEGG pathway displayed separately.
Reference:
Ding H, Chen C, Zhang T, et al. Identification of miRNA–mRNA Networks Associated with Pigeon Skeletal Muscle Development and Growth[J]. Animals, 2022, 12(19): 2509.
Transcriptional Characteristics Showed That miR-144-y/FOXO3 Participates in Embryonic Skin and Feather Follicle Development in Zhedong White Goose
Point 10: In the 3.7 change in gene expression related to sex determination, you can not only pat attention to the dmrt1 gene, but also the genes with similar expression trends to the dmrt1. It can also enrich the content of the discussion section.
Response 10: Thank you very much for your valuable suggestion. In the 3.7 change in gene expression related to sex determination, we showed the other 26 up-regulated DEGs in PMT hit against the GO terms. In addition, there were two down-regulated DEGs in PMT hit against the GO terms. (marked in red in Line 343-358)

Round 2
Reviewer 1 Report
Comments for the authors:
The resubmitted manuscript entitled “Testicular transcriptome of male and pseudo-males provides important new insight into sex reversal of Rana dybowskii” was revised; furthermore, a new figure was added. Overall, according to the reviewer’s comments, the contents were improved. However, the newly added figure has some problems.
Major point:
Figure 5 is incomplete. The label is missing for the vertical axis. The authors should represent the data of RNA-seq and qRT-PCR obtained from the testes of pseudo-males and males and compare them. Moreover, the authors should analyze them statistically.
Author Response
Response to Reviewer 1 Comments
Point 1: Figure 5 is incomplete. The label is missing for the vertical axis. The authors should represent the data of RNA-seq and qRT-PCR obtained from the testes of pseudo-males and males and compare them.
Response 1: Thank you very much for your valuable suggestion. According to the suggestions, we have remade Figure 5 (marked in red in Line 369-374).
Point 2: Moreover, the authors should analyze them statistically .
Response 2: Thank you very much for your valuable suggestion. We added the statistical analysis in line 186-189 (marked in red ).
